# Corneal Cross-Linking for Pediatric Keratoconus

**DOI:** 10.3390/diagnostics14171950

**Published:** 2024-09-03

**Authors:** Bogumiła Wójcik-Niklewska, Erita Filipek, Paweł Janik

**Affiliations:** 1Department of Pediatric Ophthalmology, Faculty of Medical Sciences in Katowice, Medical University of Silesia, 40-055 Katowice, Poland; erita.filipek@gmail.com; 2Professor Kornel Gibiński University Hospital Center, Medical University of Silesia, 40-514 Katowice, Poland; 3Institute of Biomedical Engineering, Faculty of Science and Technology, University of Silesia in Katowice, 39 Będzińska Street, 41-200 Sosnowiec, Poland; powel.janik@us.edu.pl

**Keywords:** corneal cross-linking, keratoconus, children, cornea

## Abstract

Aim: The aim of this study was to evaluate corneal cross-linking (CXL) for keratoconus in pediatric patients. Materials and methods: After keratometric qualification according to the Amsler–Krumeich system, corneal collagen cross-linking was performed using ultraviolet light and photosensitizing riboflavin drops in 111 eyes of 74 children with a mean age of 15 ± 1.67 years. None of the children studied wore contact lenses before the procedure. Visual acuity, intraocular pressure, keratometry, and pachymetry parameters were analyzed before and after corneal cross-linking. Results: Visual acuity was 0.64 ± 0.31 and 0.66 ± 0.29 before CXL and at the end of the follow-up, respectively; the difference was not statistically significant. The mean intraocular pressure before CXL was 14.48 ± 3.13 mmHg, while the mean value at the end of the follow-up was 14.23 ± 3.03 mmHg; no statistically significant difference was found. Pre- and post-CXL astigmatism was 3.98 ± 2.34 Dcyl and 3.63 ± 1.86 Dcyl, respectively; the difference was not statistically significant. The mean keratometry before CXL was 47.99 ± 3.96 D; the mean post-follow-up value was 47.74 ± 3.63 D. The mean corneal thickness (pachymetry) at the apex of the keratoconus—the thinnest zone of the cornea—before CXL was 492.16 ± 38.75 µm, while the mean value at the end of the follow-up was 479.99 ± 39.71 µm; the difference was statistically significant. Conclusions: Corneal cross-linking is an effective method for preventing keratoconus progression in children. However, further and detailed ophthalmic follow-up of patients who underwent CXL before the age of 18 is highly advisable.

## 1. Introduction

Keratoconus is a non-inflammatory, progressive degeneration in which the cornea assumes a conical shape with a cone-like protrusion secondary to stromal thinning. It is a bilateral but usually asymmetric condition. A slit-lamp examination may reveal Fleischer’s ring (57% of patients) and Vogt’s lines (44% of patients). The prevalence of keratoconus is between 0.2 and 0.04 per 100,000, while the incidence rates are 1.5 to 25 cases per 100,000 persons a year. The highest rates typically occur in 20- to 30-year-olds and Middle Eastern and Asian ethnicities [1]. Patients with keratoconus exhibit a decrease in visual acuity due to irregular astigmatism. In advanced cases, corneal scarring occurs, further worsening visual acuity. The etiology of keratoconus is still poorly understood; its development is influenced by environmental factors such as contact lens wear, chronic eye rubbing, and allergic diseases, indicating complex interactions between environmental and genetic factors [2,3,4,5].

Keratoconus most frequently develops as an isolated disorder, but it may also be associated with genetic disorders, including Down’s syndrome (10–300-fold higher prevalence), Ehlers–Danlos syndrome, anterior polar cataract, vernal keratoconjunctivitis, retinitis pigmentosa, Leber congenital amaurosis, and mitral valve prolapse [2]. It should be emphasized, though, that the genetic factors have not been clearly identified. 

Typically, keratoconus develops during puberty, with unilateral deterioration in visual acuity due to increasing irregular astigmatism and myopia. According to the CLEK study, keratoconus is diagnosed at a mean age of 27.3 ± 9.5 years, but with modern diagnostic methods, it can be detected at early stages [2]. Keratoconus is more aggressive in children than in adult patients, with rapidly progressive visual acuity deterioration requiring regular ophthalmic monitoring [6].

In the early stages of keratoconus development, spectacle correction and contact lenses can improve visual acuity, but as the condition progresses, surgical procedures are required, including penetrating keratoplasty, deep anterior lamellar keratoplasty, and intracorneal ring segment implantation. Corneal cross-linking (CXL) [7,8] is a more modern and minimally invasive procedure for patients with early and moderate keratoconus. The method uses UVA ultraviolet radiation and photosensitizing riboflavin eyedrops to induce cross-links between the collagen fibers of the corneal stroma. After treatment, the intercellular matrix is thickened, and the subepithelial nerve plexuses and keratocyte density are altered, allowing the cornea to regain some of its mechanical strength [9,10,11].

### Aim

This study aimed to evaluate the effectiveness of corneal cross-linking for keratoconus in pediatric patients.

## 2. Patients and Methods

Seventy-four children (mean age: 15 ± 1.67 years) treated in the Department of Pediatric Ophthalmology, Faculty of Medical Sciences in Katowice, Medical University of Silesia, Poland, were included in the study.

The study group consisted of 111 eyes diagnosed with progressive keratoconus stages 1 to 3 according to the Amsler–Krumeich classification. None of the children in the study had previously worn contact lenses. Before corneal cross-linking, each child underwent a complete ophthalmic examination with keratometry and pachymetric evaluation using CASIA optical coherence tomography.

After pediatric and anesthesiological qualification, the procedure was performed in all children under sedation and local anesthesia according to the Dresden protocol [11,12,13]. Cross-linking was started by removing the corneal epithelium and applying 0.1% riboflavin eyedrops for 30 min at 1–5 min intervals to achieve therapeutic concentrations in the corneal stroma. For another 30 min, the cornea was exposed to 365–370 nm UVA (irradiance of 3 mW/cm^2^) with simultaneous riboflavin instillation. The procedure was completed by inserting a contact lens, which was maintained until corneal re-epithelialization. After the procedure, antibiotic and dexpanthenol eyedrops were prescribed. Following epithelium regrowth, decreasing doses of steroid drops were administered for another three weeks. A comprehensive ophthalmic examination was performed during follow-up appointments, including keratometry and pachymetry.

The degree of astigmatism and keratometry and pachymetry readings were analyzed before and after corneal cross-linking. The follow-up period was 2.5 years.

The conformity of the data distributions to a normal distribution was verified by the Shapiro–Wilk test and assessed graphically using histograms. A repeated-measures ANOVA or its non-parametric equivalent, the Friedman rank test, was used to compare the results obtained before CXL (preCXL), one week after (postCXL-1), and at the end of the follow-up (postCXL-2). Statistical significance was set at *p* < 0.05. The results were graphically presented using the estimated margins and box-and-whisker plots. Descriptive statistics (arithmetic means and standard deviations) were used to summarize and describe the characteristics of each group.

## 3. Results

The parameters subjected to statistical analysis were evaluated before CXL, one week after the procedure, and at the end of the follow-up period.

The mean visual acuity was as follows: pre-CXL: 0.64 ± 0.31; one week after the procedure: 0.65 ± 0.30; at the end of the observation period: 0.66 ± 0.29. Due to the left-skewed distributions, the Friedman test was used. There was no statistically significant difference between the measurements (*p* = 0.2904) (Figure 1).

The mean intraocular pressure before CXL was 14.48 ± 3.13 mmHg; one week after the procedure and at the end of the follow-up, the values were 14.07 ± 2.95 and 14.23 ± 3.03 mmHg, respectively. There was no statistically significant difference between the measurements (repeated-measures ANOVA, *p* = 0.2027) (Figure 2).

The mean pre-CXL corneal thickness (pachymetry) at the apex of the keratoconus (the cornea’s thinnest zone) was 492.16 ± 38.75 µm. One week after the procedure and at the end of follow-up, the mean values were 485.56 ± 39.62 and 479.99 ± 39.71 µm, respectively. The intermeasurement difference was statistically significant (repeated-measures ANOVA, *p* = 0.0002) (Figure 3). Tukey’s post hoc test showed a significant difference between PachyApex-preCXL and PachyApex-postCXL-2. No significant difference was revealed between PachyApex-postCXL-1 and PachyApex-postCXL-2 (*p* = 0.0673).

The mean pre-CXL astigmatism was 3.98 ± 2.34 Dcyl; the respective values at one week after CXL and at the end of the follow-up were 3.72 ± 1.86 Dcyl and 3.63 ± 1.86. The difference was not statistically significant (Friedman test, *p* = 0.9209) (Figure 4).

The mean pre-CXL keratometry was 47.99 ± 3.96 D; the respective values at one week after CXL and at the end of the follow-up were 47.95 ± 3.97 D and 47.74 ± 3.63 D. The difference was not statistically significant (Friedman test, *p* = 0.06345) (Figure 5).

No inflammatory complications were observed in any child during the entire follow-up period, and none of the eyes required a repeat CXL or penetrating keratoplasty.

## 4. Discussion

The development of modern diagnostic methods to identify early-stage keratoconus and numerous reports showing positive outcomes of corneal cross-linking have contributed to this procedure being widely accepted for treating the disease [9,11,14,15]. Cross-linking of corneal collagen fibers has been shown to significantly inhibit keratoconus progression. Many authors have confirmed its high effectiveness. The results of a meta-analysis by Meiri et al. showed an improvement in visual acuity after CXL and a reduction in astigmatism by 0.4 to 0.7 diopters. Pachymetry values decreased by 10 to 20 µm in the year after the procedure but not after 24 months [11].

Treating pediatric patients with keratoconus is especially challenging, as the disease progresses rapidly [4,11,12]. Corneal cross-linking has been suggested as the first treatment to be used when diagnosing disease progression in these patients. Soeters et al. [16] performed CXL in patients with confirmed keratoconus progression within the preceding 1 to 3 months (pediatric cases) and 6 to 12 months (adolescents and adults). Chatzis and Hafezi take a different view, immediately qualifying children with a family history of keratoconus or presenting with rapid progression of keratoconus in the fellow eye. The authors concluded that corneal cross-linking seemed to be safe for children and adolescents. However, they also believe that the stopping of disease progression might not be as long-lasting as in adults and that a longer follow-up is needed to verify this trend [17]. In contrast, Magli et al. [18] and Shetty et al. [19] recommend observing children for six months before performing corneal cross-linking. In our Pediatric Ophthalmology Department, we take the position that once a child is diagnosed with keratoconus, immediate qualification for corneal cross-linking is required. We have observed that failure to perform this procedure promptly is associated with rapid disease progression and deterioration in visual acuity.

Eligibility criteria for children with keratoconus are the same as for adult patients and include corneal thickness of at least 400 µm and absence of opacities, vernal keratoconjunctivitis, viral infections, and other corneal infections. Exclusion criteria are autoimmune diseases, severe dry eye syndrome, previous ocular surgery, and less than 1000 endothelial cells per mm^2^ [4,20,21]. These were the criteria we used in our study. 

Vinciguerra et al. studied 40 eyes of patients below 18 years; the follow-up period was 24 months. They found improved visual acuity and stabilization of keratometry readings, resulting from a reduction in astigmatism and higher-order aberrations [22]. We studied 111 eyes of 74 children after corneal cross-linking. Similar to Vinciguerra et al., we did not find deterioration in visual acuity after 2.5 years of follow-up. Caporossi et al. analyzed corneal topography parameters of 152 children aged 10 to 18 who had undergone CXL for progressive keratoconus. At 36 months, visual acuity and topography readings showed statistically significant improvement [23]. In another study with 48 months of minimum follow-up, the authors detected keratoconus stability in 44 eyes. The mean K value decreased by two diopters [24]. The post-CXL keratometry readings of our patients did not differ significantly compared to pre-CXL values. 

The lack of deterioration in our patients’ visual acuity is essential from the point of view of the children’s schooling. The stabilization of the local condition, and thus the absence of the need for invasive procedures involving long-term immunosuppressive medications, was also an essential benefit for both the children and their parents. 

Most authors report the presence of some inflammatory complications after CXL. Fung et al. studied 21 eyes of 13 subjects aged ≤18 years. During a median follow-up of 14.5 months, they found a sterile corneal infiltrate in one eye, which resolved with a short course of corticosteroids [25]. Varshney et al. reported a case of an 11-year-old boy who developed post-CXL acute corneal hydrops. This complication was managed conservatively; the condition of the cornea was monitored with anterior segment optical coherence tomography. Corneal hydrops resolved in 3 weeks, which was accounted for by increased interlamellar cohesive strength between collagen fibrils of corneal stroma coupled with a normal function of the endothelial pump [26]. We have found no such complications.

## 5. Conclusions

Corneal cross-linking is an effective method of preventing keratoconus progression in children. However, further and detailed ophthalmic follow-up of patients who underwent CXL before the age of 18 is advisable.

## Figures and Tables

**Figure 1 diagnostics-14-01950-f001:**
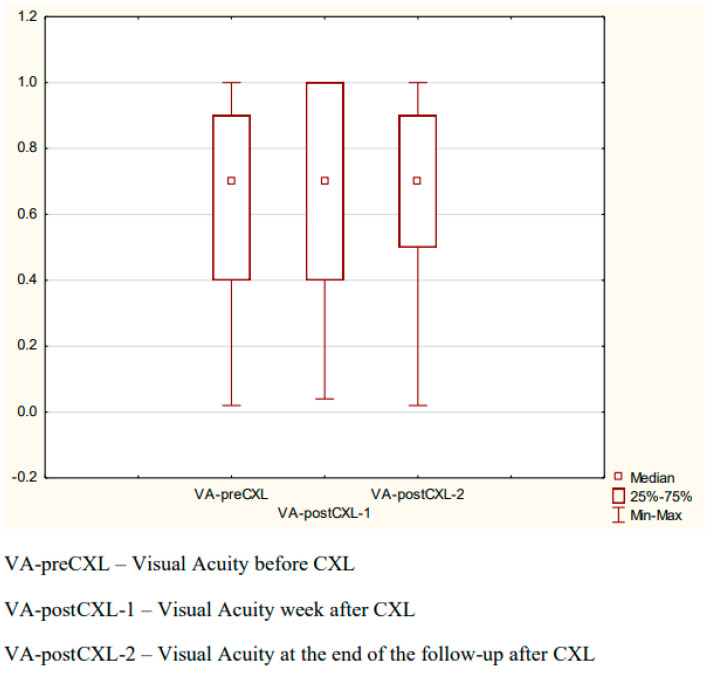
Visual acuity before and after CXL.

**Figure 2 diagnostics-14-01950-f002:**
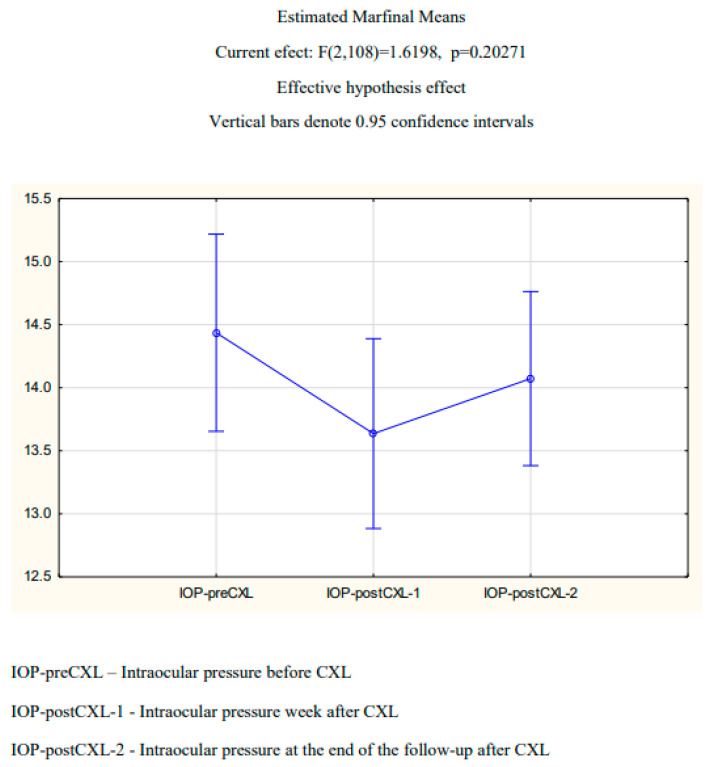
Intraocular pressure before and after CXL.

**Figure 3 diagnostics-14-01950-f003:**
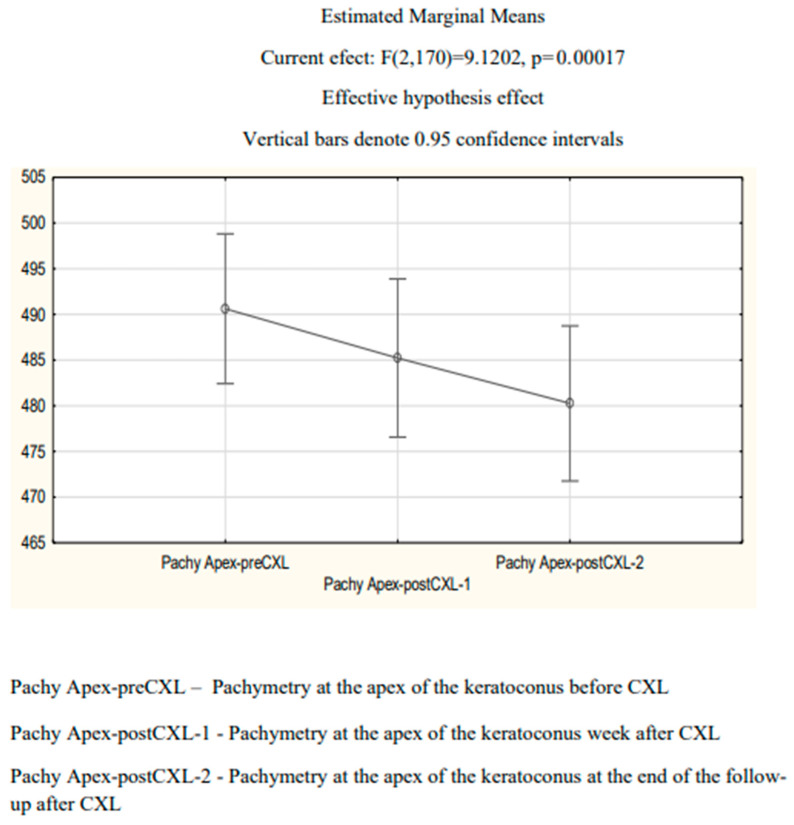
Mean pachymetry at the apex of the keratoconus (µm) before and after CXL.

**Figure 4 diagnostics-14-01950-f004:**
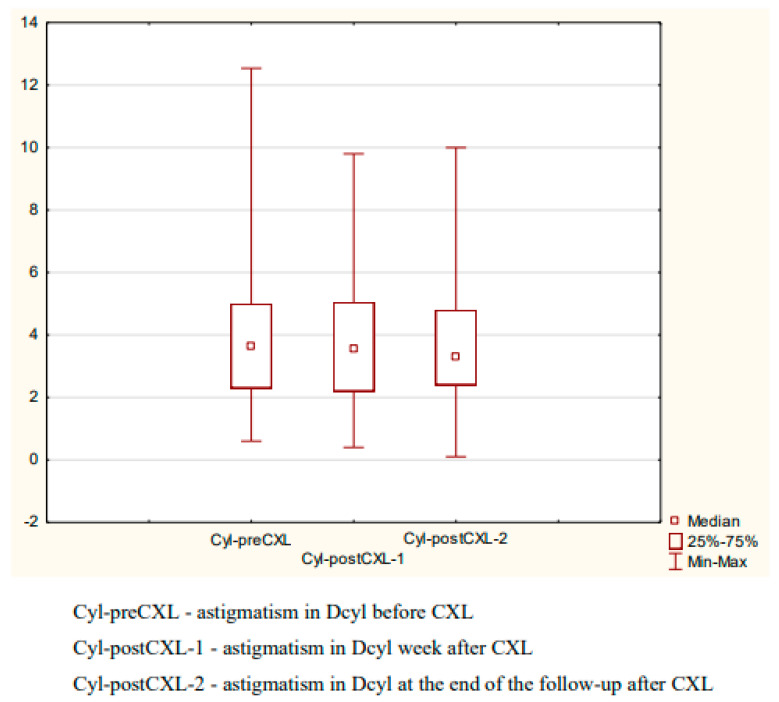
Astigmatism before and after CXL.

**Figure 5 diagnostics-14-01950-f005:**
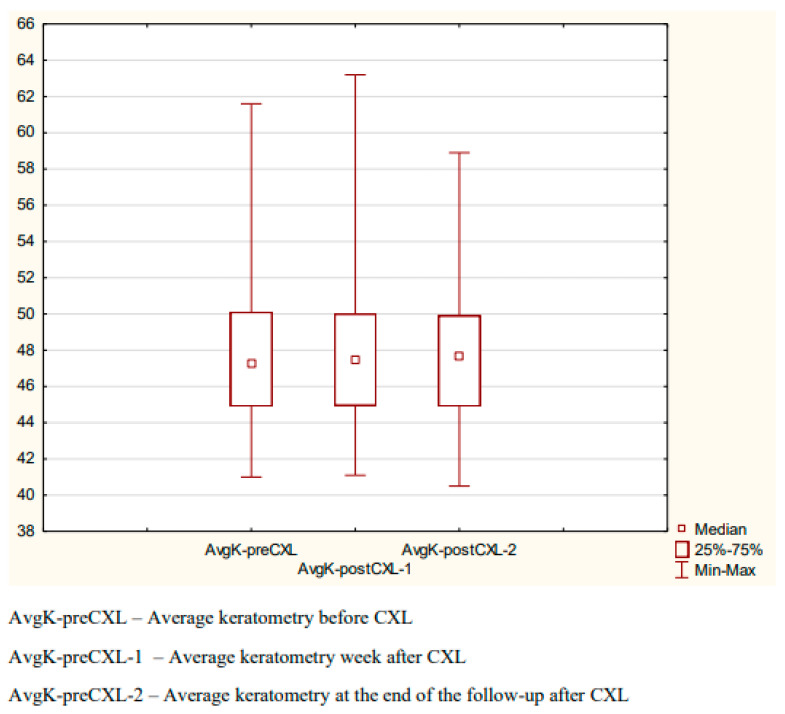
Mean keratometry before and after CXL.

## Data Availability

The data presented in this study are available on request from the corresponding author. The data are not publicly available due to General Data Protection Regulation.

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
