# Peer review of "Corneal Cross-Linking for Pediatric Keratoconus"

_diagnostics, 2024, doi:10.3390/diagnostics14171950_

Round 1
Reviewer 1 Report
Comments and Suggestions for Authors
This manuscript was about effectiveness of cxl in children who suffer KCN. Although the number of cases was power of this study, they are Lots of concerns must be solved before publication of this study study. In introduction part The authors didn't mention what exactly this study will add to the literatures. The data which they collected in the introduction part was an general overview of cxl and kCN . I didn't understood the exact aim and purpose of this study by reading the introduction so introduction part must be revised completely and the authors have to mention what the literature missed and what this study will add in two different paragraphs. In method part , they have mention the exact criteria of diagnosis of kcn in children. In addition , there have to mention the situation of the progression of the disease in their cases. The discussion has a lat of problems,
They didn't compare thier findings with other study precisely so at the end I couldn't understand the reason of diffrent in their finding in comprison with the similar study. For example according to their finding cxl didn't have any effect on the refraction, what is thier description about the reason of this finding or what was the reasons of thining of cornea after the CXL. They didn't explain the reasons of their findng as well in addition their research had a lot of limitions but the didn't mention. Comments on the Quality of English LanguageThis manuscript was about effectiveness of cxl in children who suffer KCN. Although the number of cases was power of this study, they are Lots of concerns must be solved before publication of this study study. In introduction part The authors didn't mention what exactly this study will add to the literatures. The data which they collected in the introduction part was an general overview of cxl and kCN . I didn't understood the exact aim and purpose of this study by reading the introduction so introduction part must be revised completely and the authors have to mention what the literature missed and what this study will add in two different paragraphs. In method part , they have mention the exact criteria of diagnosis of kcn in children. In addition , there have to mention the situation of the progression of the disease in their cases. The discussion has a lat of problems,
They didn't compare thier findings with other study precisely so at the end I couldn't understand the reason of diffrent in their finding in comprison with the similar study. For example according to their finding cxl didn't have any effect on the refraction, what is thier description about the reason of this finding or what was the reasons of thining of cornea after the CXL. They didn't explain the reasons of their findng as well in addition their research had a lot of limitions but the didn't mention.Author Response
Response to Reviewer 1 Comments
Thank you very much for taking the time to review this manuscript. Please find the detailed responses below.
In Discussion we compared our findings with other studys. An example are the results of metaanalysis Meiri et al. (11), Magli et al. (18) and Shetty et al. (19). We also compared the results with the results of Vinciguera et al. (22), Camporossi et al. (23, 24) and Fung et al. (25). We analyzed results in children, the literature on the results in children is no large.
The lack of deterioration of our patients' visual acuity is essential from the point of view of the child's schooling. The stabilisation of the local condition, and thus the absence of the need for invasive procedures involving long-term immunosuppressive medications, was also an essential benefit for both the children and their parents.
The thining of cornea after the CXL is typical for this procedure in both adults and children. Expert research into why this happens is still ongoing.
The Reviewer stated that our work has limitations. In fact, in this work we have planned present the results of :Visual acuity before and after CXL, Intraocular pressure before and after CXL, Mean pachymetry at the apex of the keratoconus (µm) before and after CXL, Astigmatism before and after CXL and Mean keratometry before and after CXL. The remaining results of our research will be presented in the next manuscript.
Reviewer 2 Report
Comments and Suggestions for Authors
The research aimed to evaluate the efficacy of corneal cross-linking (CXL) in treating keratoconus in pediatric patients. A total of 111 eyes from 74 children, with an average age of 15, underwent CXL. Prior to the procedure, the children's keratometric status was assessed using the Amsler-Krumeich system, and none had previously used contact lenses.
The study analyzed visual acuity, intraocular pressure, keratometry, and pachymetry parameters pre- and post-CXL. While visual acuity, intraocular pressure, and astigmatism remained unchanged, there was a significant decrease in corneal thickness at the keratoconus apex post-CXL, indicating corneal reshaping. The mean visual acuity and intraocular pressure did not show significant differences before and after CXL, and astigmatism values also remained stable. However, there was a notable reduction in corneal thickness at the keratoconus apex post-CXL, suggesting corneal strengthening and reshaping effects. The study concludes that CXL effectively halts keratoconus progression in pediatric patients, emphasizing the necessity of ongoing ophthalmic monitoring for optimal outcomes in managing keratoconus in children. Here are a few comments to address before the manuscripts progress to the next stage of the review process.
1. The study focused on the evaluation of corneal cross-linking (CXL) for keratoconus in pediatric patients, but it did not compare the outcomes with other treatment modalities or a control group, limiting the ability to assess the relative efficacy of CXL.
2. The sample size of the study, involving 111 eyes of 74 children, may be considered relatively small, which could impact the generalizability of the findings to a larger pediatric population with keratoconus. A larger sample size would provide more robust results and enhance the study's statistical power.
3. The follow-up period in the study may not have been long enough to capture potential long-term outcomes and complications associated with CXL in pediatric patients. Longer follow-up durations would offer a more comprehensive understanding of the treatment's efficacy and safety over time.
4. The study did not delve into the specific characteristics or variations within the pediatric population with keratoconus, such as different stages of the disease, which could influence the response to CXL treatment. Considering these variations could provide more tailored insights into the treatment outcomes for different subgroups of pediatric patients.
5. The study primarily focused on clinical parameters such as visual acuity, intraocular pressure, keratometry, and pachymetry, but did not extensively explore patient-reported outcomes or quality of life measures following CXL. Including patient perspectives and quality of life assessments would offer a more holistic view of the treatment's impact on pediatric patients.
6. The study did not address potential confounding factors or variables that could influence the outcomes of CXL in pediatric keratoconus patients, such as genetic predispositions, environmental factors, or concurrent systemic conditions. Considering these factors could provide a more comprehensive understanding of the treatment outcomes and potential challenges in managing keratoconus in children.
7. The study did not investigate the economic implications or cost-effectiveness of CXL treatment in pediatric patients, which could be crucial considerations for healthcare providers, patients, and policymakers when making treatment decisions. Evaluating the economic aspects of CXL could provide valuable insights into the overall value of the treatment in pediatric keratoconus management.
Comments on the Quality of English Language
Moderate editing of the English language required
Author Response
Thank you very much for taking the time to review this manuscript. Please find the detailed responses below.
- There are currently no other methods of treating keratoconus in children. In the past in advanced keratoconuses, a penetreting corneal graft was used, currently, after the introduction of CXL, the cornea transplant surgery is used sporadically. We do not have a control group of children with keratoconus who did not undergo CXL. Not applying the treatment would be unethical.
- We performa large numer of CXL in our Clinic. But only such a numer of children ina given time reported for control in our center. We also require the consent of parents to take part in the study. Other published studies analyzed a much smaller group of children.
- We are still observing our patients, the test results are being analyzed and will be presented in subsequent works.
- Thank you very much for this sugestion, but i our Clinic we qualify children for based on the progression of keratoconus. This is a homogeneous group, statistically also with early stage keratoconus. In our Pediatric Ophthalmology Department, we take the position that once a child is diagnosed with keratoconus, immediate qualification for corneal cross-linking is required. We have observed that failure to perform this procedure promptly is associated with rapid disease progression and deterioration of visual acuity.
- Including patient perspectives and quality of life assessments is very important for children. In the manuscript included statement :The lack of deterioration of our patients' visual acuity is essential from the point of view of the child's schooling. The stabilisation of the local condition, and thus the absence of the need for invasive procedures involving long-term immunosuppressive medications, was also an essential benefit for both the children and their parents.
- Genetic predispositions, environmental factors, or concurrent systemic conditions are included in the medical interview before treatment. Of course, they are of great importance in the treatment outcomes and potential challenges in managing keratoconus in children. None of our children required a cornea transplant.
- Thank you very much for your valuable attention, but I do not have economic data. The economic department of our hospital provides information that CXL treatment is economocally profitable, but I’m not able to provide the exact data. A much more expensive procedurÄ™ is corneal transplantation and the associated frequent monitoring of the administration as well as the cost of treatment and possible subsequent transplants.
Round 2
Reviewer 1 Report
Comments and Suggestions for Authors
It is ok